# Defeat, Entrapment, and Hopelessness: Clarifying Interrelationships between Suicidogenic Constructs

**DOI:** 10.3390/ijerph191710518

**Published:** 2022-08-24

**Authors:** D. Nicolas Oakey-Frost, Emma H. Moscardini, Kirsten Russell, Susan Rasmussen, Robert J. Cramer, Raymond P. Tucker

**Affiliations:** 1Department of Psychology, Louisiana State University, 236 Audubon Hall, Baton Rouge, LA 70803, USA; 2School of Psychological Sciences and Health, University of Strathclyde, 40 George Street, Glasgow G1 1QE, UK; 3Department of Public Health Sciences, University of North Carolina at Charlotte, 9201 University City Blvd., Charlotte, NC 28223, USA

**Keywords:** defeat, entrapment, hopelessness, suicidal ideation, factor structure, theory

## Abstract

Psychological theories of suicide posit conceptually similar constructs related to the development of suicidal thinking. These constructs often evince high-magnitude interrelationships across studies. Within these theories, defeat, entrapment and hopelessness standout as conceptually and quantitatively similar. Theoretical improvements may be facilitated through clarifying the subscale and item-level similarities among these constructs. Factor analytic and phenomenological work has demonstrated equivocal evidence for a distinction between defeat and entrapment; hopelessness is not typically analyzed together with defeat and entrapment despite evidence of large-magnitude interrelationships. This study explored the interrelationships among the foregoing constructs within a sample of undergraduate students (*N* = 344) from two universities within the Southeastern United States. Participants, oversampled for lifetime history of suicidal ideation and attempts, completed an online cross-sectional survey assessing defeat, entrapment, hopelessness and SI. Exploratory factor and parallel analyses demonstrated support for a one factor solution when analyzed at subscale level of the three measures as well as when all items of the three measures were analyzed together. Ad hoc exploratory structural equation modeling (ESEM) bifactor results evinced support for the existence of a single, general factor at the item level. Item level communalities and bifactor fit indices suggest that hopelessness may be somewhat distinct from defeat and entrapment. Clinical and theoretical implications are discussed in the context of study limitations.

## 1. Introduction

In a seminal review addressing the, sometimes, fuzzy operational boundaries within contemporary suicidology, Silverman and colleagues [1] clarified operational definitions of key constructs within suicide prevention research. Concise conceptualization of constructs and behaviors yields a research body that is communicable across fields of practice from the macro-level down to the individual. While this review focused specifically on suicidal thought and behavior (STB) terminology, clarity and parsimony are especially applicable to prevailing theoretical work and to understanding the dynamics of risk and resiliency for STB [2]. The proliferation of these fuzzy operational boundaries may be the function of an absence of a common theoretical framework that concisely defines and applies suicidogenic constructs toward successful treatment and prevention of STB [3]. Thus, more empirical work is needed to clarify individual interpretation, internalization, and relationships among the myriad suicidogenic constructs.

Of the constructs commonly thought to confer risk for STB in contemporary psychological theories for suicide, hopelessness, defeat, and entrapment stand out as conceptually and quantitatively similar. Hopelessness is defined as negative expectations for the future and appears to be a central cognitive mechanism for STB vulnerability [4]. Over several decades, empirical work has demonstrated a reliable relationship between hopelessness, suicide attempts [5], and suicidal ideation (SI) [6]. Additionally, hopelessness has been shown to predict suicide attempts over both a 10-year period and shorter follow-up intervals (i.e., baseline to 6-month, 24-month, and 48-month follow-up) [7,8,9].

Defeat and entrapment were originally conceptualized as the key components of the Cry of Pain Model of suicide and self-harm [10]. They are, respectively, defined as a sense of failed struggle and losing rank and escape motivation triggered by either external or internal cues [11]. Both constructs share relationships with a host of psychiatric diagnoses and STBs [12,13,14]. Nevertheless, their phenomenological and clinical utility is muddied by a body of psychometric evidence suggesting significant conceptual and quantitative overlap calling into question their unique construct validity [15,16,17].

Defeat and entrapment may be measured using a variety of self-report instruments; these include the Personal Beliefs about Illness Questionnaire (PBIQ) [18], semi-structured clinical interviews of mental defeat and attachment [19], the pain self-perception scale (PSPS) [20], the Involuntary Submission Questionnaire (ISQ) [21], and the Caregiver Burden Scale-Entrapment Subscale (CBS-E) [22]. Nevertheless, the Defeat Scale (DS) and the Entrapment Scale (ES) [11] are the two most commonly used measures within the suicidology literature [13] and are thus used to conceptualize these constructs within the current investigation. Similarly, hopelessness may be measured via several valid instruments [23,24], but the Beck Hopelessness Scale (BHS) [25] remains the most commonly used within the suicidology literature to date.

Item-level observation of the self-report measures used to assess hopelessness, defeat, and entrapment yields examples of the conceptual similarities. DS items include statements such as *“I feel powerless”,* conceptually similar to *“Things just don’t work out the way I want them to”* from the BHS [11,25]. Both items evoke the definition of defeat (i.e., a sense of failed struggle) as necessary to preclude a desired outcome yet exist within seemingly divergent measures. Similarly, entrapment and hopelessness share conceptual overlap in items such as *“My future seems dark to me”* from the BHS and *“I feel I am in a deep hole I can’t get out of*” from the ES [11,25].

The conceptual similarity between defeat and entrapment has a storied history within the broader psychopathology literature to date [14]. For example, previous attempts at deriving measures of immediate escape from traumatic experiences included concurrent conceptualization of mental planning for escape (i.e., entrapment) and later mentally relinquishing a fight response (i.e., defeat) to interpersonal trauma [19,20]. In one case in particular, these parallel conceptualizations of defeat and entrapment were observed to load on to a single, latent factor [20]. Accordingly, Griffiths and colleagues [16] among others [14,26] propose that defeat and entrapment comprise a single factor, often emerge from a single event, and co-occur to form a “depressogenic loop” in particular circumstances, rendering them borderline indistinguishable. Thus, it is difficult to determine how patients or participants may distinguish between these constructs and what that means for prevailing suicidology research and clinical application. Further research that attempts to determine if these constructs are either (1) distinct yet highly related or (2) redundant may be enlightening.

Despite the item-level communalities between questionnaires designed to measure these purportedly distinct constructs, little empirical work has focused on differentiating hopelessness from defeat and entrapment [16]. The available empirical examples appear to demonstrate that the relationship between hopelessness and entrapment consistently demonstrate bivariate correlation coefficients that range from *r =* 0.70–0.83 and hopelessness with defeat from *r* = 0.64–0.81, evincing strong convergence at a minimum [12,27]. Despite these strong relationships, no available scientific literature appears to examine the interrelationship between the three constructs beyond bivariate correlation analyses.

The scientific debate surrounding the convergence of defeat and entrapment is much more robust; in fact, the available psychometric evidence for the relationship between defeat and entrapment is arguably inconclusive [15]. Siddaway and colleagues [13] conclude that entrapment is a particularly relevant cognitive risk mechanism for STB, but the relationship between entrapment and STB demonstrate only a face-value average difference from those of defeat and STB, not a *statistically significant* difference, suggesting that the two constructs may indeed represent “two sides of the same medal” [15] (p. 6). In a multi-method comparison, Forkmann and colleagues [15] used network, exploratory, and confirmatory factor analytic methods within both clinical and online samples, observing that a one factor solution of the DS and ES [11] is preferred over a three or four factor solution (i.e., defeat, internal, external entrapment, and a three item “winner” cluster). Forkmann and colleagues [15] initially contend that their findings are consistent with those of Griffiths and colleagues [28] where the two constructs were found to represent a unidimensional factor, later replicated in development of the short defeat and entrapment scale (SDES) [16]. Notably, Griffiths and colleagues [16] observed superior fit indices in favor of a two-factor solution; however, the two factors were also observed to correlated at a magnitude of *r* = 0.91, suggesting redundancy. Indeed, earlier evidence pointed toward a similar unidimensional solution accounting for nearly 50% of the variance among all items of the defeat and entrapment scales [26].

Nevertheless, Forkmann and colleagues [15] ultimately conclude that defeat and entrapment should be interpreted considering clinical utility and theoretical fit, contending that their data indicate two distinct constructs as interpreted through the lens of Social Rank Theory [11]. By contrast, Höller and colleagues [17] found that a two-factor solution of the DS and ES outperformed a one-factor solution in two of three samples and the final pooled sample, suggesting that defeat and entrapment are best conceptualized as originally intended within the Social Rank theory of depression [11,29]. In sum, the psychometric research to date appears equivocal at best, and none of these investigations have considered the conceptual and quantitative overlap of defeat and entrapment together with hopelessness.

Within the broader context of theoretical and applied suicidology, the foregoing measures are typically leveraged in research capacity to inform phenomenological assertions based in theory (e.g., integrated motivational-volitional model) [2]. However, most “verbal”, or language-based theories of suicide leverage purportedly distinct constructs which demonstrate at minimum non-zero interrelationships and are often moderate to high in magnitude [3]. Thus, the applied utility of distinguishing between these constructs (both clinically and in research) remains debatable [30].

This study aimed to add to ongoing theoretical debate via exploration of the interrelationships between hopelessness, defeat, and entrapment as measured by the BHS, DS, and ES. Extensive research has investigated bivariate relationships between these constructs (e.g., defeat and entrapment) [15], but none have further investigated these simple relationships utilizing psychometric procedures. Pooling items across multiple self-report scales and performing psychometric analyses has provided clarity on the conceptual overlap of interrelated psychological constructs [31]. Thus, the current investigation examines item-level similarities and differences across the BHS, DS, and ES to further inform construct validity. While correlations between defeat, entrapment, and hopelessness were hypothesized to be positive in direction and large in magnitude, no hypotheses were made regarding psychometric analyses given the exploratory nature of this study.

## 2. Method

**Participants.** The sample pool (*N* = 344) consists of United States (US) adults recruited from two large state universities. The first sample consisted of *n* = 210 students recruited from the student research participant pool of a large, south-central US university. This sample consists of predominantly white (76.19%) women (78.10%), aged 18 to 54 (*M* = 19.82, *SD* = 3.97). The second sample consisted of *n* = 134 students recruited from the student research participant pool of a different large, southern US university. This sample consists of predominantly white (60.44%) women (70.90%) aged 17 to 29 (*M* = 19.14, *SD* = 1.77). See Table 1 for the demographic breakdown of each sample.

**Procedures.** Individuals in both samples were recruited via oversampling procedures based on responses to the first item of the Suicide Behaviors Questionnaire (SBQ-R) [32]. Respondents who endorsed a history of at least one suicide attempt or a history of SI received an email invitation to participate. For both data collections, emails sent to participants included a brief description of the study and an anonymous web link redirecting them to the survey collection software online domain where they were asked to provide informed consent and complete questionnaires. After survey completion, participants were asked to read debriefing information including local and national crisis and counseling service contact information. More information regarding study procedures can be found elsewhere [33].

## 3. Materials

**Demographics.** The survey asked participants to indicate their age, gender identity, and racial/ethnic identity.

**Hopelessness.** A modified, validated version The Beck Hopelessness Scale (BHS) [25,34] was used to measure hopelessness. The BHS was originally designed to measure negative expectancies for the future [25]. In contrast to the original [25], asking participants to answer a series of twenty prompts in a dichotomous true/false response format, the modified version [34] asks participants to respond to the same prompts on a five-point Likert-type scale from zero to four. The modified version of the BHS was chosen to have all items across these scales rated on Likert-type scales to match the response pattern of other measures to aid in item level psychometric analyses. Iliceto and Fino [34] indicate that three first-order factors (i.e., BHS Motivation, BHS Affect, BHS Cognitive) and one second order factor solution represents the best fit for the data. In this sample, the three first order factors were used for subscale analyses representing Affective, Motivation, and Cognitive domains of hopelessness. Within the current sample, all the Affective, Motivation, and Cognitive sub-scales demonstrated adequate to good internal consistency, α = 0.79–0.89 while the BHS total score demonstrated excellent internal consistency, α = 0.94.

**Defeat.** The Defeat Scale (DS) [11] assesses perceptions of defeat and demoralization. Participants are asked to read a series of 16 statements and rate the extent to which each statement reflects how they have felt about themselves over the last seven days on a five-point Likert-type scale from zero to four where higher scores indicate greater feelings of defeat. Historically, the DS demonstrates excellent internal consistency, α = 0.93–0.94 [11]. Within the current sample, the DS similarly demonstrated excellent internal consistency, α = 0.96.

**Entrapment.** The Entrapment Scale (ES) [11] measures beliefs that one is trapped by external and internal situations. Originally validated with a two-factor structure of internal and external entrapment, the internal subscale measures motivation to escape internal thoughts and feelings while the external subscale measures motivations to escape aversive environmental stimuli [11]. Participants are asked to read a series of 16 statements and rate the extent to which each statement represents their view of themselves on a five-point Likert-type scale from zero to four. Historically, the ES has demonstrated good to excellent internal consistency for both the internal (α = 0.82–0.94) and external (α = 0.86–0.90) subscales [11]. Within the current sample, both the external entrapment and internal entrapment subscales demonstrated excellent internal consistency, α = 0.93–0.95, respectively. Internal consistency for the ES overall was excellent, α = 0.96.

## 4. Analytical Plan

Both samples were collapsed into a single, larger, pooled sample to sufficiently power all planned analyses. Despite identical recruitment procedures, the samples significantly differed based on constructs of interest with those in Sample 2 demonstrating higher mean scores on all measures (see Online Appendix A). Bivariate correlation analyses were conducted to determine simple relationships between study variables at the construct or subscale level. Exploratory factor analytic (EFA) techniques were utilized to further test statistical overlap between the constructs. In the first EFA, relationships were analyzed at the construct or subscale level. Thus, subscale scores of the BHS (i.e., BHS Motivation, BHS Affect, BHS Cognitive), ES (internal and external entrapment), and DS served as the observed variables. In the second EFA, all items across all three measures were entered across measures as used previously [31]. In each of these analyses, principal axis factoring with direct oblimin rotation was utilized as factors extracted were expected to correlate but may demonstrate some distinction. Parallel analysis [35] was used to determine how many factors should be extracted. All analyses were conducted in SPSS [36].

## 5. Results

### 5.1. Correlational Findings

Table 2 includes means, standard deviations, and bivariate correlations between defeat, entrapment, hopelessness, and their subcomponents. All bivariate associations were positive in direction and moderate to large in effect size.

### 5.2. Exploratory Factor Analysis at the Construct Level

The Kaiser-Meyer-Olkin (KMO) sampling adequacy statistic was 0.88 and the Bartlett’s test of sphericity was significant (*χ*^2^(15) = 1715.36, *p* < 0.001) indicating sufficient factorability. Only one factor had an extracted eigen value above 1.0; this factor (eigen value = 4.40) explained 73.38% of the variance. A parallel analysis also indicated a one-factor solution with the following eigen values: factor 1 = 4.40, factor two = 0.69. All total scores/subscale scores loaded on the single factor above 0.71.

### 5.3. Exploratory Factor Analysis at the Item Level

No items across the DS, ES, or BHS demonstrated initial communalities below 0.30. The Kaiser-Meyer-Olkin (KMO) sampling adequacy statistic was 0.97 and Bartlett’s test of sphericity was significant (*χ*^2^(1275) = 15,420.72, *p* < 0.001) indicating sufficient factorability. A parallel analysis indicated a four-factor solution with the following eigen values: factor one = 24.21 (46.56% of the variance explained), factor two = 3.40 (6.54% of the variance explained), factor three = 1.9 (3.66% of the variance explained), factor four = 1.28 (2.46% of the variance explained), factor five = 0.95 (1.83% of the variance explained). The parallel analysis was repeated with a fixed four factor solution given the results of the parallel analysis. See Online Appendix A for all item factor loadings. In this repeated analysis with four factors, the majority of the items correlated with the first factor with multiple significant cross-loadings. No distinct, observable pattern of loadings was seen, but generally items that loaded most strongly on the first factor included 23 items all forward coded from the DS and ES across both subscales. Items that loaded most strongly on the second factor included 11 items all forward coded items across all BHS subscales. The third factor included 12 items all reverses coded from BHS and the DS (i.e., a method factor). The fourth factor included 6 items all forward coded from the ES across subscales.

### 5.4. Ad Hoc Analysis

The item level EFA data evinced traits of both unidimensionality and multidimensionality. Specifically, the data indicate the possibility of a one-factor solution as the optimal solution; the first factor’s eigen value was 7.12 times the value of the subsequent factors and there was significant item cross-loading observed. A factor structure of this nature is observed in the current study and past work with the DS and ES [16,28] suggestive of a single, common latent trait and that all items within the respective questionnaires measure this common latent trait [37]. Further, the significant cross-loading of items suggest a violation of the independent cluster assumption for exploratory factor models [38]. The significant cross-loading of items between factor solutions indicates that items of the DS, ES, and BHS tap into similar content domains and that a more parsimonious solution may best fit the available response data [37]. One solution that has been proposed to account for substantial evidence of possible unidimmensionality between scales and observed multidimensionality among factor solutions is an exploratory bifactor approach [39]. Thus, the EFA at the item-level across all three measures was repeated using an exploratory structural equation model (ESEM) bifactor approach, which allows for cross loading of items to determine the extent to which each scale should be represented by a single latent factor [40]. We conducted four ESEM bifactor analyses with two to five specific factors.

Within the bifactor approach, goodness of fit was determined using indices for Root Mean Square Error of Approximation (RMSEA), Standardized Root Mean Square Residual (SRMR), Chi-square, Tucker–Lewis Index (TLI), and Comparative Fit Index (CFI). The following cutoffs were used to evaluate model fit: RMSEA < 0.06, SRMR < 0.08, TLI > 0.95, CFI > 0.95, respectively, which would demarcate “good” or “excellent” fit [41]. To determine if the set of items should be interpreted as a unidimensional construct within the bifactor solutions, percent of uncontaminated correlations (PUC), percent of explained common variance (ECV), item-level explained common variance (IECV), and omega hierarchical (ωH) were calculated. The PUC estimate is the percentage of covariance terms which are reflective of variance from only the general dimension, and common variance is considered unidimensional when PUC > 0.70 and ECV > 0.70 [42]. The IECV estimate reflects the percentage of item level variance accounted for by a unidimensional latent factor which can then be used to choose items to create a more unidimensional scale [42,43]. IECV values > 0.80 are recommended for creating a suitable unidimensional scale [44]. Coefficients of ωH indicate the percentage of systematic variance in raw total scores attributable to individual differences on the general factor [42,45]. Coefficients of ωH that are >0.80 are considered unidimensional [42,45]. To compare the inferred bifactor solutions, changes in CFI, RMSEA and TLI > 0.01 were considered significant [46].

Factor determinacy (FD) and construct reliability were also calculated where values > 0.90 suggest that bifactor subscales are sufficient for use [47] and construct replicability values > 0.80 suggest a well-defined latent construct [48,49]. Average relative parameter was calculated to determine the difference between the item’s loading from the unidimensional solution and general factor loading from the bifactor solution [42,45]. Values below 0.015 provide support for unidimensionality [42,45].

### 5.5. Ad Hoc Analysis Results

Global fit statistics for the bifactor analyses are seen in Table 3. The exploratory bifactor model with two specific factors [*χ*^2^(3266) = 42,068.98 *, *p* < 0.05, TLI = 0.955, CFI = 0.960, RMSEA = 0.063, SRMR = 0.046] demonstrated good fit, meeting most predetermined criteria. Although the ESEM bifactor models with three through five inferred specific factors evinced slightly superior in fit, we continued with the bifactor model with two specific factors considering the generally higher item loadings when compared with the other inferred solutions. We continued the analyses of whether this scale is largely influenced by a general factor. Of the 52 items, 48 loaded loaded significantly onto the general factor, all of which loaded onto the general factor with values > 0.39.

IECV values are reported in Table 4. More than half of the items (31) had IECV values > 0.8 or 0.85, indicating that these 31 items yield a unidimensional item set; of note, *all items within this set* included defeat and entrapment items [44]. All items that did not have IECV values > 0.80 derived from the BHS across all subscales, while two BHS items evinced IECV values > 0.80. Nevertheless, the following estimates supported a unidimensional factor interpretation: percent of explained common variance (ECV; 0.82), PUC (1.0), omega hierarchical (*ωH* = 0.98), overall relative parameter bias (0.05) [42,45] FD for the general factor and specific factors 1 and 2 were >0.95, indicating that these factor score estimates are suitable for use as a subscale.

## 6. Discussion

The aim of this study was to explore and examine the interrelationship between hopelessness, defeat, and entrapment at both the construct/subscale and item levels. At the construct level, results of the EFA demonstrate a clear best fit for a one factor solution, arguably converging with and expanding on the results of Forkmann and colleagues [15]. At the construct level, the current findings converge with previous work; only a single factor eigen value exceeded those generated by chance in EFA and follow-up parallel analysis [16,26,28]. Indeed, the first inferred factor within the EFA was estimated to explain nearly 75% of variance in response to items within the dataset. Additionally, it is telling that the best fitting and most parsimonious solution of the ad hoc exploratory bifactor analysis are nearly identical to that of the EFA and associated parallel analysis; a single defeat/entrapment factor, a specific hopelessness factor, and a methods factor. Thus, at the subscale level, the evidence potentially suggests that defeat and entrapment may best be measured and operationalized as a single construct contrary to previous assertion [15]. As indicated by the current results, this inferred model may now include the construct of hopelessness.

However, the current findings diverge with those of Höller and colleagues [17] who found that a two-factor solution of the German version of the Short Defeat and Entrapment Scale (SDES) [16] outperformed a one factor solution within an online, an inpatient, and the final aggregate sample. Nevertheless, Höller and colleagues [17] observe and comment on objectively mixed findings where both the one factor and two factor solutions evince only marginal differences in fit. Additionally, they argue that distinguishing between these is clinically practical yet observed that patient self-rated defeat and entrapment scores did not differ between participants reporting SI vs. those who did not, and those participants who had a history of attempting suicide vs. those with no suicide attempt history [17]. Thus, it is unclear as to whether distinguishing between these constructs is clinically practical based on the current findings and those of Höller and colleagues [17]. Indeed, those who experience STBs appear to experience a complicated amalgamation of defeat, entrapment, and hopelessness as we currently understand them, and no available language-based descriptor appears to appropriately capture these basic cognitive and emotional processes [3].

The observed results of the ad hoc ESEM bifactor analysis at the item level suggest that, at a minimum, all items within the ES and DS may comprise a unidimensional set and that response to these items are accounted for by variation in the general dimension alone given that the majority of their estimated IECV values fall above 0.85 [43]. Upon observation of the item level loadings, the two specific factors are comprised solely of BHS items (factor 1) and reverse scored items from all scales in question (factor 2), strongly implicating the existence of a methods factor. These observations provide support for two potential conclusions: (1) defeat and entrapment may indeed comprise “two sides of the same medal” as argued by Forkmann and colleagues [15] and (2) if a specific factor does indeed exist, it is characterized by hopeless cognitions only. Thus, while the majority of the BHS items comprising the specific factor loaded more strongly on to the general factor, there may yet be a quantifiable, and thus a theoretical and phenomenological distinction between defeat/entrapment and hopelessness that should be examined further.

The convergence of these oft studied suicidogenic constructs into a single factor is not a novel observation. Bryan and Harris [50] demonstrated that the suicidogenic constructs (e.g., hopelessness, defeat, entrapment) captured within the Suicide Cognitions Scale (SCS) may be best conceptualized as a “family” of related cognitions that characterizes the more general and *singular* suicidal belief system within suicide-related theories such as the fluid vulnerability theory (FVT) [51,52]. Indeed, the current findings align with those of Bryan and Harris [50]; a network of highly interrelated constructs is characteristic of a more general suicidal belief system that is not entirely distinguishable as hopelessness, defeat, nor entrapment specifically.

Of note, the authors do not propose that these constructs (i.e., hopelessness, defeat, entrapment) be used to create a novel amalgamated scale at this stage; rather, like those of the SCS-R, the results suggest the existence of a “suicidal belief system” which “reflects a distorted and maladaptive thinking style” where no unique set of suicidogenic cognitions is more or less important in potentiating suicidal behavior [53] (p. 10). Thus, the authors are cautious to label the observed unidimensional general factor considering recent theoretical discussion [3], but the observed results may indeed support the optimally parsimonious operationalization of the suicidal belief system [1,50,52].

Finally, the results suggest that individuals may not be able to conceptually distinguish between these constructs at the level of language; as Millner and colleagues [3] argue, “most clinical constructs have non-zero intercorrelations” (p. 3) as observed in the current study, and these verbal, language based constructs may do little to advance our knowledge of STB phenomenology, especially if populations in questions cannot or do not distinguish between or respond differentially to these constructs. Thus, more generalized concepts and phenomena such as the suicidal belief system become more and more appealing to theoretical and practical understanding as the field of suicidology advances in understanding [54].

## 7. Limitations

The results should be interpreted considering several important limitations which will hopefully guide future research. First, the sample size used in this study was relatively small, thus future work would benefit by replicating results in a larger sample. The sample size limitation prohibits more robust assertions as to whether these constructs truly capture the same dimension or whether there are simply not enough participants for the related but distinct constructs to dissociate from the general factor. Additionally, numerous instruments are used to conceptualize these constructs across a number of different populations [13]. Considering these foregoing concerns, replication of the current results is needed to more confidently assert the unidimensional nature of these oft-studied constructs. The sample is comprised primarily of young adult White cisgender female undergraduate students, and thus the results of this study may not be generalizable to other populations. Collapsing two independent samples of participants from two different universities was needed to increase statistical power, but this methodological limitation too limits generalizability of study findings. Finally, this study only investigated concurrent relationships between hopelessness, defeat, and entrapment which could conceal important clinically relevant temporal variations between these constructs and STB. Case in point, Abramson and colleagues [55] argue that hopelessness is comprised of both state- and trait-like characteristics, potentially explaining why some variance in hopelessness was observed as primarily defining the general dimension in conjunction with defeat and entrapment, and additional specific and independent variance. Of course, these are only preliminary phenomenological assertions; however, longitudinal examining the criterion validity of *composite* defeat/entrapment/hopelessness scores as compared to temporally segregating these constructs would be incredibly enlightening.

## 8. Conclusions

The findings from both studies indicate considerable conceptual and statistical overlap between defeat, entrapment, and hopelessness. Results from the EFA and bifactor analyses indicate that items from the BHS, ES, and DS may best be interpreted as influenced by one latent factor instead of three distinct factors. This assertion appears to apply to all potential subscales between these measures. Future research would benefit from further investigations into this possible distinction so as to improve theories regarding SI risk and provide important information for clinicians assessing risk for SI with patients. Empirical investigations of whether constructs such as hopelessness, entrapment, and defeat are meaningfully distinct are needed to inform parsimonious suicide risk assessments and interventions. Although unidimensionality between these constructs and their measures may point to the need to simply “collapse” them into one larger construct of interest, differential predictive validity of these factors/measures may point to the need to keep them separate.

## Figures and Tables

**Table 1 ijerph-19-10518-t001:** Demographic data for study samples 1 and 2 (*N* = 344).

	Sample 1 (*n* = 210)	Sample 2 (*n* = 134)
Demographics		
Age	*M* = 19.81 (*SD* = 3.97)	*M* = 19.14 (*SD* = 1.77)
*Race/Ethnicity*		
% White	160 (76.19%)	81 (60.45%)
% Black/African American	9 (4.35%)	27 (20.15%)
% Asian/Asian-American	6 (2.86%)	7 (5.22%)
% Latino(a)(Latinx)	2 (0.95%)	1 (0.75%)
% Indigenous	11 (5.24%)	12 (8.96%)
% Biracial/Multiracial	21 (10%)	1 (0.75%)
% Other	1 (4.76%)	5 (3.73%)
*Gender*		
% Woman	164 (78.10)	95 (70.90%)
% Man	44 (20.95%)	29 (21.64%)
% Not listed	NA	10 (7.5%)

Note. Percentages may not equal 100 as participants could select more than one response.

**Table 2 ijerph-19-10518-t002:** Means, Standard Deviations, and Correlation Coefficients of Defeat, Entrapment, and Hopelessness.

Variable	1	2	3	4	5	6	7
1. Defeat Scale	-						
2. External Entrapment	0.81 *	-					
3. Internal Entrapment	0.85 *	0.83 *	-				
4. BHS Total	0.74 *	0.67 *	0.68 *	-			
5. BHS Affective	0.71 *	0.64 *	0.64 *	0.88 *	-		
6. BHS Motivation	0.59 *	0.54 *	0.53 *	0.89 *	0.62 *	-	
7. BHS Cognitive	0.77 *	0.62 *	0.65 *	0.91 *	0.73 *	0.74 *	-
*M*	23.08	11.58	7.94	24.07	8.29	6.77	9.01
*SD*	14.51	10.20	7.64	15.67	5.87	6.38	5.31
*Skew*	0.406	0.597	0.628	0.624	0.417	1.090	0.469
*Kurtosis*	−0.660	−0.674	−0.901	−0.338	−0.817	0.521	−0.466

Note: * *p* < 0.001.

**Table 3 ijerph-19-10518-t003:** Global fit indices for factor analyses.

Model	df	*χ* ^2^	CFI	TLI	SRMR	RMSEA
ESEM Bifactor with two specific factors	1326	42,068.98 *	0.960	0.955	0.046	0.063
ESEM Bifactor with three specific factors	1124	2195.06 *	0.974	0.969	0.037	0.053
ESEM Bifactor with four specific factors	1076	1853.81 *	0.981	0.976	0.031	0.046
ESEM Bifactor with five specific factors	1029	1610.64 *	0.986	0.982	0.027	0.041

Note. df = degrees of freedom; RMSEA = root mean square error approximation; CFI = comparative fit index; SRMR = standardized root mean residual; * = *p* < 0.05.

**Table 4 ijerph-19-10518-t004:** Item-factor loadings and IECV from the ESEM bifactor analysis of the Defeat Scale, the Entrapment Scale, and Beck’s Hopelessness Scale.

	Gen	1	2	IECV
The Defeat Scale				
DS1	0.790	−0.106	−0.032	0.981
DS2	0.645	−0.111	0.334	0.771
DS3	0.858	−0.101	−0.050	0.983
DS4	0.672	−0.062	0.335	0.796
DS5	0.832	−0.030	−0.149	0.968
DS6	0.777	−0.031	−0.199	0.937
DS7	0.830	−0.077	−0.078	0.983
DS8	0.840	−0.086	−0.123	0.969
DS9	0.527	−0.049	0.270	0.787
DS10	0.878	0.072	−0.229	0.930
DS11	0.854	0.061	−0.291	0.892
DS12	0.849	0.143	−0.154	0.942
DS13	0.838	−0.033	−0.081	0.989
DS14	0.827	−0.088	−0.136	0.963
DS15	0.812	−0.002	−0.199	0.943
DS16	0.811	0.079	−0.0117	0.971
The Entrapment Scale				
ES1	0.860	−0.077	−0.070	0.986
ES2	0.851	−0.173	−0.011	0.960
ES3	0.491	0.073	−0.229	0.807
ES4	0.819	−0.204	−0.052	0.938
ES5	0.859	−0.058	−0.042	0.993
ES6	0.794	−0.160	−0.092	0.949
ES7	0.814	0.060	−0.096	0.981
ES8	0.684	−0.012	−0.248	0.884
ES9	0.770	−0.128	−0.056	0.968
ES10	0.79	−0.060	−0.227	0.908
ES11	0.858	−0.229	0.067	0.928
ES12	0.832	−0.084	−0.005	0.990
ES13	0.777	−0.279	0.004	0.910
ES14	0.830	−0.285	0.031	0.906
ES15	0.840	−0.224	0.019	0.935
ES16	0.878	−0.024	−0.047	0.997
Beck’s Hopelessness Scale				
BHS1	0.801	−0.031	0.339	0.847
BHS2	0.646	0.448	0.083	0.668
BHS3	0.543	−0.022	0.340	0.718
BHS4	0.398	0.280	0.025	0.667
BHS5	0.643	−0.008	0.234	0.883
BHS6	0.693	0.081	0.475	0.674
BHS7	0.705	0.453	0.123	0.693
BHS8	0.550	−0.106	0.366	0.676
BHS9	0.513	0.520	−0.117	0.481
BHS10	0.552	0.006	0.347	0.717
BHS11	0.657	0.535	0.080	0.596
BHS12	0.605	0.481	−0.022	0.612
BHS13	0.364	0.121	0.518	0.319
BHS14	0.677	0.509	−0.125	0.625
BHS15	0.774	0.066	0.461	0.734
BHS16	0.593	0.716	−0.037	0.369
BHS17	0.693	0.697	0.046	0.419
BHS18	0.644	0.398	0.144	0.698
BHS19	0.723	0.023	0.394	0.770
BHS20	0.529	0.717	0.001	0.352

Note: IECV = item explained common variance, DS = Defeat Scale, ES = Entrapment Scale, BHS = Beck Hopelessness Scale.

## Data Availability

Not applicable.

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
