# Peer review of "Defeat, Entrapment, and Hopelessness: Clarifying Interrelationships between Suicidogenic Constructs"

_ijerph, 2022, doi:10.3390/ijerph191710518_

Round 1

Reviewer 1 Report

The authors presented and original study on the interrelationship between psychological constructs involved in the development of suicidality. The topic of the study is of clinical, research and preventative importance. The manuscript is well written, the rationale and the aims are clearly stated. The methodology and analysis appear sound a rigorous. A very minor observation or recommendation - the authors may want to consider to include some reference on the work of J.M.G. Williams, who was one the first to illuminate and study defeat, entrapment and hopelessness in relation to suicidality. Some suggestions:

Williams, J.M.G. (1997). Cry of pain: Understanding suicide and self-harm. London: Penguin.

Williams, J.M.G. & Pollock, L. (2001). Psychological aspects of the suicidal process. V C. van Heeringen (Eds.)), Understanding suicidal behavior: the suicidal process approach to research, treatment and prevention (pg. 76-94). Chichester: Wiley

Other than that the paper is of high quality and the authors really are to be congratulated for this contribution.

Author Response

Reviewer 1

The authors presented and original study on the interrelationship between psychological constructs involved in the development of suicidality. The topic of the study is of clinical, research and preventative importance. The manuscript is well written, the rationale and the aims are clearly stated. The methodology and analysis appear sound a rigorous. A very minor observation or recommendation - the authors may want to consider to include some reference on the work of J.M.G. Williams, who was one the first to illuminate and study defeat, entrapment and hopelessness in relation to suicidality. Some suggestions:

Williams, J.M.G. (1997). Cry of pain: Understanding suicide and self-harm. London: Penguin.

Williams, J.M.G. & Pollock, L. (2001). Psychological aspects of the suicidal process. V C. van Heeringen (Eds.)), Understanding suicidal behavior: the suicidal process approach to research, treatment and prevention (pg. 76-94). Chichester: Wiley

Thank you for this feedback. A reference to Williams & Pollock (2001) has been added to the revised manuscript and can be found on page 2 of 19. It reads:

Defeat and entrapment were originally conceptualized as the key components of the Cry of Pain Model of suicide and self-harm [10]. They are, respectively, defined as a sense of failed struggle and losing rank and escape motivation triggered by either external or internal cues [11].

Other than that the paper is of high quality and the authors really are to be congratulated for this contribution.

Reviewer 2 Report

Thanks for assessing the convergence of the interrelated factors. Overall, it is a well-written paper. I have some suggestions.

1. Authors are requested to mention all the scales measuring Defeat, entrapment, and hopelessness with justifications for using the three specific scales in this study. Do the authors think that changing scales would reveal a different observation?

2. I could find any clue regarding the justification of the study in the abstract.

3. Methods: Ethical approval details could be erased from the section and placed at the bottom section of the manuscript. 

4. I am not sure whether IJERPH has moved to format-free submission. Authors are requested to read the submission guidelines and follow especially the referencing. 

Author Response

Reviewer 2

Thanks for assessing the convergence of the interrelated factors. Overall, it is a well-written paper. I have some suggestions.

  1. Authors are requested to mention all the scales measuring Defeat, entrapment, and hopelessness with justifications for using the three specific scales in this study. Do the authors think that changing scales would reveal a different observation?

Thank you for this feedback. The authors have now added a brief mention of other instruments used to measure defeat and entrapment, specifically. The addition can be found on page 2 of 19 of the revised manuscript and reads as follows:

Defeat and entrapment may be measured using a variety of self-report instruments; these include the Personal Beliefs about Illness Questionnaire (PBIQ) [18], semi-structured clinical interviews of mental defeat and attachment [19], the pain self-perception scale (PSPS) [20], the Involuntary Submission Questionnaire (ISQ) [21], and the Caregiver Burden Scale-Entrapment Subscale (CBS-E) [22]. Nevertheless, the Defeat Scale (DS) and the Entrapment Scale (ES) [11] are the two most commonly used measures within the suicidology literature [13] and are thus used to conceptualize these constructs within the current investigation. Similarly, hopelessness may be measured via several valid instruments [23, 24], but the Beck Hopelessness Scale (BHS) [25] remains the most commonly used within the suicidology literature to date.

Additionally, although the authors cannot speculate as to whether the results would replicate using alternative measurement instruments, we have added this as a limitation of the study on page 14 of 19 of the revised manuscript. It reads as follows:

Additionally, numerous instruments are used to conceptualize these constructs across a number of different populations [13]. Considering these foregoing concerns, replication of the current results is needed to more confidently assert the unidimensional nature of these oft-studied constructs.

  1. I could find any clue regarding the justification of the study in the abstract.

Thank you for this feedback. The authors have added a sentence to the abstract which we hope communicates he general intention of the investigation. The abstract now reads as follows:

Abstract: Psychological theories of suicide posit conceptually similar constructs related to the development of suicidal thinking. These constructs often evince high magnitude interrelationships across studies. Within these theories, defeat, entrapment and hopelessness standout as conceptually and quantitatively similar. Theoretical improvement may be facilitated through clarifying the subscale and item-level similarities among these constructs. Factor analytic and phenomenological work has demonstrated equivocal evidence for a distinction between defeat and entrapment; hopelessness is not typically analyzed together with defeat and entrapment despite evidence of large magnitude interrelationships. This study explored the interrelationships among the foregoing constructs within a sample of undergraduate students (N=344) from two universities within the Southeastern United States. Participants, oversampled for lifetime history of suicidal ideation and attempts, completed an online cross-sectional survey assessing defeat, entrapment, hopelessness and SI. Exploratory factor and parallel analyses demonstrated support for a one factor solution when analyzed at subscale level of the three measures as well as when all items of the three measures were analyzed together. Ad-hoc exploratory structural equation modelling (ESEM) bifactor results evinced support for the existence of a single, general factor at the item level. Item level communalities and bifactor fit indices suggest that hopelessness may be somewhat distinct from defeat and entrapment. Clinical and theoretical implications are discussed in the context of study limitations.

  1. Methods: Ethical approval details could be erased from the section and placed at the bottom section of the manuscript. 

Ethical approval details have been removed from the Methods section and can be found at the end of the manuscript, as requested.

  1. I am not sure whether IJERPH has moved to format-free submission. Authors are requested to read the submission guidelines and follow especially the referencing. 

Thank you for this feedback. The authors have changed the references in text and in the Reference list to match IJERPH style.